# Improving outcomes for primary school children at risk of cerebral visual impairments (the CVI project): study protocol for the process evaluation of a feasibility cluster-randomised controlled trial

Anna Pease ![ORCID], Trudy Goodenough, Parisa Sinai, Katie Breheny, Rose Watanabe, Cathy Williams ![ORCID]

► http://dx.doi.org/10.1136/bmjopen-2020-044830

Bristol Medical School, University of Bristol, Bristol, UK

**Correspondence to**
Dr Cathy Williams;
cathy.williams@bristol.ac.uk

## ABSTRACT

**Introduction** Brain-related visual impairments, also known as cerebral visual impairment (CVI), are related to damage or poor function in the vision-related areas of the brain. There is broad agreement that CVI is an appropriate term to describe visual impairments that are not accounted by disorders of the eye or optic nerve, but differences remain as to which impairments can be included in this term. The CVI project is a programme of work that includes the development of a complex intervention to share knowledge with teachers, so that they can make both targeted and universal changes to support children with CVI. A feasibility study for a cluster-randomised controlled trial to evaluate this intervention is underway. This paper describes the protocol for an accompanying process evaluation to explore how the intervention is implemented and provide context for the interpretation of the feasibility trial outcomes.

**Methods and analysis** A logic model has been developed to guide data collection. Both qualitative and quantitative data will be collected to assess the feasibility and acceptability of the intervention, the study design and explore how any changes that occur are brought about. Interviews with key primary school staff and parents will investigate responses to the intervention and trial processes. Surveys will collect data on intervention implementation and knowledge of CVI. Photographs of classroom walls will document any changes to visual clutter and document analysis will look for changes to school special educational needs and disability (SEND) policies.

**Ethics and dissemination** Ethical approval was granted by the University of Bristol Faculty of Health Sciences Ethics Committee. Findings will contribute to the development of a full-scale cluster-randomised controlled trial to assess the effectiveness of the intervention with adequate statistical power. The results will also support the refinement of the intervention and its underlying theory.

### Strengths and limitations of this study

► Intervention developed with input from professionals and families.
► Novel intervention to improve awareness of cerebral visual impairment and strategies to help.
► Process evaluation designed to inform future full-scale trial, if warranted.
► Study designed using programme logic to inform data collection timelines and methods.
► Ethical approval re-established following COVD-19-related pause in the study.

## INTRODUCTION

Damage or poor function in the brain areas related to vision leads to brain-related visual impairments, also known collectively as cerebral visual impairment (CVI).[1 2] CVI in children can manifest in many ways; for example, being unable to find things on a cluttered page; bumping into people; being unable to copy from the class whiteboard to their own workbooks or difficulties controlling their eye position effectively to keep focused on a task.[3] It has been reported that vision problems may be misinterpreted by teachers as behavioural, learning or intellectual difficulties[4] and so children with CVI may not get the help they need. Concern has been raised that missing a diagnosis of CVI can affect mental health,[5 6] and research from a UK birth cohort suggested that children with CVI may experience academic difficulties.[7] Research into strategies to support children with CVI has suggested that providing a less cluttered visual environment may help.[8 9]

In preparation for this study, discussions with two parent advisory groups, and with

an advisory group comprising local health and education professionals, have reinforced accounts that simple adaptations of the child's visual world can be very helpful. They emphasise children's improved sense of well-being when they can achieve more and are not chastised for behaviour resulting from their CVI. They have told us that access to help is uneven between and within different areas of the country, and, therefore, is inequitable, and that few teachers, special educational needs co-ordinators (SENCO) or learning support assistants know about CVI. In general, only children with reduced visual acuity have access to specialist vision support teachers and so many children with CVI but with good visual acuity are ineligible for their help. A prevalence study conducted by our group has indicated that there are likely to be at least 3% of children with CVI-related vision problems in primary schools in England and that these are largely but not exclusively within the subset of children already identified as needing extra educational help.[10]

### Aim of the process evaluation

The aim of the CVI project process evaluation (PE) is to understand both the functioning of the trial processes and the intervention itself. This understanding is crucial to the development and decision to proceed to the full trial.

## METHODS AND ANALYSIS
### Patient and public involvement

We have spoken with two groups of parents whose children have special educational needs and/or CVI. They have been consulted at every stage of the research so far and provided guidance on which measures to use with parents and which topics to include in qualitative interviews. During the study design phase, the parents advised us on the need to raise awareness in schools about CVI, and we changed our draft intervention pack to include specific resources to address this. They were clear that teachers should be reimbursed for their time to take part in the study and that we should outline the potential benefits of identifying CVI, whereby the right strategies can be put in place, freeing up teacher time not spent trying ineffective interventions. They also advised us that schools might appreciate the benefit of being able to demonstrate to the Office for Standards in Education inspectors that they were implementing strategies to support students who may be struggling with their learning.

### The CVI project intervention

We have developed an intervention pack that seeks to share knowledge with teachers so that they can make both universal and targeted changes within their school, to better support children with CVI and potentially other children too.[9]

The development of the intervention pack is based on recent guidance, including a review of the published literature, using an iterative process of development

and refining, involving key stakeholders, developing an underlying theory that takes into account the context and possible real-world implementation issues.[11] The key elements of the intervention and the hypothesised outcomes are summarised in a logic model—see figure 1.

We hypothesise that this multi-faceted approach, involving both schools and the hospital service, will improve educational and wellbeing-related outcomes for any child with CVI and potentially for their peers as well. We have, therefore, devised an information sharing pack to give schools this information. This includes an overview and explanations of the condition and gives suggestions as to how to make the school more 'vision friendly'. Central to this is to use universal approaches wherever possible—for instance, decluttering walls and classrooms, using larger font and clear layouts with worksheets. Additional targeted strategies, for use with at-risk children, will also be given, with explanations about who might benefit from these, how they are intended to work and their potential impact. The information included in the intervention pack for teachers is collated from several centres of excellence that have developed materials about CVI, including Bartiméus in the Netherlands[12] and the international TeachCVI initiative (www.Teachcvi,net).[13] The intervention also includes sharing information with the local paediatric ophthalmology clinics about the presentations of and support for children with CVI, so that they are happy to accept referrals of at-risk children. Intervention schools will be able to refer a maximum 5% of children to the paediatric ophthalmology clinic for a functional visual CVI assessment. A printout of the CVI assessment will be available to the family and school with strategies where applicable.

A logic model showing proposed mechanisms of change, impact and outcomes is shown in figure 1. This model has been refined throughout our intervention development with parents and professionals and includes consideration of unintended negative consequences of the intervention, or 'dark logic', as proposed by Bonell and colleagues.[14]

### The CVI project feasibility trial

The intervention and trial processes are being tested in a cluster-randomised feasibility trial, with the nested process and health economic evaluations. The main aim of this trial is to assess the feasibility and acceptability of our intervention and trial methods, to inform the development of a future adequately powered cluster-randomised controlled trial (RCT), if warranted.[15] The full protocol for the feasibility trial is presented separately and is summarised here, to give context for the PE, which is described in detail below.

The feasibility trial involves mainstream primary schools in three regions of the UK: Gloucester, Hampshire and Somerset. Baseline and follow-up data will be collected on students recruited in years 3–5 attending participating schools.

**Figure 1** CVI project intervention logic model. CVI, cerebral visual impairment; DVD, digital video disc; GP, General Practitioner; FIM, Family Impact Model; PEDSQoL, Pediatric Quality of Life Inventory TM; QoL, Quality of Life.

In designing the study, we were aware of two challenges; first, that in real life it would be impractical to examine all children to look for CVI and second that even if those were possible, once identified, we would need to help the affected children and would not, therefore, have a control group. However, in our recent study,[10] we found that most children with CVI were found within the group of children getting extra help at school, and these children are easily identifiable by schools. In this feasibility trial, therefore, the primary outcome (child self-reported mental well-being) will be assessed in all children who are getting extra help, rather than in only children with confirmed CVI. We will not assess every child in the study for CVI.[10] In addition, we also found some children with CVI-related vision problems who were not getting extra help, and we hypothesise many children might benefit from some aspects of the intervention, for example, decluttering the classroom to make the visual environment less busy. We will, therefore, collect data for *all* children in years 3–5. The primary outcome of child-reported quality of life will be assessed in the children getting extra help, and we will include all children in an additional (secondary) analysis using the same outcome.

After baseline data are collected, a statistician separate from the study will randomise the participating schools, with two stratification variables to maintain similar sample sizes in each arm[16] (1) 15% or less of children

having extra help or more than 15% of children and (2) size of school (three more classes per year vs two or fewer classes per year, in years 3–5). Schools will be told immediately whether they are allocated to the control group (and will be asked to carry on as normal for the school year) or the intervention group (where they will be invited to implement the CVI intervention pack, described above).

Some of the outcomes of the feasibility study will be assessed using data collected during the study in a PE. The PE will use a mixture of quantitative and qualitative methods for data collection. Complex interventions are described as those which contain several interacting components and may also relate to the implementation of the intervention and its interaction with its context.[17] PEs embedded within complex interventions aim to understand the functioning of an intervention by examining implementation, mechanisms of impact and contextual factors.[18] There are several frameworks that combine these aspects of PE within feasibility of complex interventions and RCTs; for this study, our framework is based on guidance from the Medical Research Council (MRC).[17] We have used this guidance to develop the aims and conceptual framework for this PE. These are described below.

The seven domains to be evaluated as recommended by MRC guidance are as follows:

## Implementation of the intervention

This domain aims to discover what is implemented in practice. We will consider how the CVI Project Intervention Training package is used in each school. We have made suggestions for how schools may wish to implement the intervention (including delivering the presentation at a staff meeting and sharing the resources with class teachers); however, we have not been prescriptive about this. Early discussions with teachers in our advisory groups suggested that individual schools would prefer to find their own ways to implement the intervention pack, and that autonomy over this process would be appreciated. We will assess the quality (fidelity) of the intervention and how much of the intervention (dose) was delivered and to whom (reach). We will also determine whether there have been any adaptations to the intervention and the reason or purpose of these adaptations. Consideration will also be given to the core components of the intervention, and the barriers and facilitators that influence how individual components of the intervention are selected and delivered.

## Mechanism of impact/change

The PE will consider how the intervention, as delivered by schools, worked. Research questions will provide information and insights into how the participants, schools and parents responded to and interacted with the intervention. Questions will examine changes in the understanding and knowledge of staff about CVI and how this interacted with their implementation of the intervention, along with exploring the barriers and facilitators they experienced to implementation. Questions will also consider if there were any unintended or unexpected consequences or pathways of the intervention.

## Programme differentiation and usual practice

The research questions in this domain will examine how the intervention differed from usual practice in the intervention schools. Contamination from intervention to control schools is a potential risk, where schools in the control group make changes related to CVI after hearing about the intervention from a school in the intervention arm. We will seek to assess this with exploratory questions about how and why this occurred and the outcome for control schools.

## Acceptability of the intervention

This domain will consider the interactions of participants, both school staff and parents with the intervention. The effects of acceptability on implementation of the intervention will be explored.

We will consider whether acceptability changes across the school year, and how the responses and interactions of staff and parents with the intervention might change the way the intervention is implemented.

## Sustainability

This domain will consider whether the intervention could become part of normal practice after the feasibility trial was completed. We will assess the fidelity, dose and reach of the intervention and any steps taken to make it sustainable. Where it has not been sustained, or there are no plans to continue with the changes after the trial, we will explore why this is. Barriers and facilitators to sustainability will also be explored.

## Context

Contextual factors may influence the effectiveness of an intervention both directly and indirectly. These are external factors that can impede or strengthen its implementation or effects (MRC).

We will consider how the paediatric ophthalmology clinics respond and interact with the intervention, including the referral pathway from schools to Hospital Eye Service (HES). In addition, this domain will consider the wider contextual factors that could affect the implementation of the intervention.

## Trial process

Understanding the participants' responses to and interactions with the trial processes will allow an assessment of how the trial methods might require adjustment or redesign for further work. Understanding the responses of the participants will also provide information about recruitment and continued participation in the trial by both schools, teachers and parents.

## Design

The CVI project is a feasibility cluster-randomised study with a nested PE and health economic assessment. The PE will use both quantitative and qualitative data sources in order to answer our research questions. We anticipate that all schools will contribute data to the PE, allowing for different experiences with the study between intervention and control arms.

## Data sources

We have summarised the research domains, research questions and methods of evaluation in table 1.

## Surveys

Each intervention school will complete a brief school-level survey during the intervention period, conducted at the beginning of the qualitative interview. The survey includes six questions about the implementation of the intervention pack including how it was delivered, who delivered it, which components of the intervention pack have been used, who received the training, how many referral letters have been sent out and how many students have been identified for referral. Follow-up questions will include numbers who have received the intervention, which components have been used, numbers of children identified and referred.

Teachers' knowledge of CVI will be assessed before randomisation and at follow-up, using the Bartiméus Centre (Netherlands) Teacher CVI Knowledge Assessment Survey.[12] All teachers for years 3–5 in all schools will complete these. The survey is comprised of five statements

**Table 1** Process evaluation research domains and methods of evaluation

| Domain | Research questions<br>*process evaluation framework mapping* | Informant/source of data | Methods | Timing of data collection | Analysis |
|---|---|---|---|---|---|
| Implementation of intervention (the information pack) Intervention schools n=4 | Which CVI project information pack components were delivered? *Fidelity, dose, reach* | SENCO/head teacher n=4 | Brief survey question | Postintervention | Data recorded and tabulated |
| | Which resources were used? | SENCO/head teacher n=4 | Brief survey questions | Postintervention | Data recorded and tabulated |
| | How were these resources used? *Fidelity* | | Semistructured interview | | Thematic analysis |
| | How was the training delivered? For example: training day, short sessions. *Fidelity, adaptation* | SENCO/head teacher n=4 | Brief survey questions | Postintervention | Data recorded and tabulated |
| | Who received the training? *Reach* | SENCO/head teacher n=4 | Brief survey questions | Postintervention | Data recorded and tabulated |
| | Were adaptations made to resources? How did this affect implementation? *Adaptation, Fidelity* | SENCO/head teacher n=4 | Semistructured interview | Postintervention | Thematic analysis |
| | What were the barriers and facilitators to delivering the training? *Mediators* | SENCO/head teacher n=4 | Semistructured interview | Postintervention | Thematic analysis |
| Mechanisms of change of CVI project intervention Intervention schools n=4 | What was the response of staff to the training? *Participant responses to and interactions with the CVI project intervention* | SENCOs/head teacher classroom teachers, LSA years 3–5 n=TBC | Semistructured interview | Postintervention | Thematic analysis |
| | How did staff knowledge about CVI change in response to training/intervention? *Participant responses, reach* | SENCO/head teacher classroom teachers, LSAs years 3–5 n=TBC | Bartiméus Centre (Netherlands) Teacher CVI Knowledge Assessment Survey | Brief online survey prerandomisation follow-up data collection. | Quantitative analysis |
| | What were staff and parents' responses to the implementation of the CVI project intervention? *Participant responses to and interactions with the CVI project intervention* | SENCO/head teacher classroom teachers, LSA years 3–5 n=TBC | Semistructured interview | Postintervention | Thematic analysis |
| | What changes to school environment took place? Eg, Whole school/class/individual child changes | SENCO/head teacher, classroom teachers, years 3–5 n=23 classrooms | Photographs taken of study designated viewpoints of each wall in years 3–5 classrooms | Prerandomisation and postrandomisation Pre-follow-up data collection | Quantitative analysis using validated University of Bristol 'clutter' measure |
| | *Fidelity, adaptations, and reach* Context of implementation of intervention | SENCO/head teacher n=4 | Semistructured interviews | Post-follow-up data collection | Thematic analysis |
| | Were there any unintended/unexpected consequences from the intervention? What were they? How could they be mitigated (if necessary)? *Unintended/unexpected pathways or consequences* | SENCO/head teacher, classroom teachers, LSAs years 3–5; parents n=TBC | Semistructured Interview | Postintervention | Thematic analysis |
| | What were the barriers and facilitators for the implementation of the CVI project intervention? *Participant responses to and interactions with the CVI project intervention* | SENCO/head teacher, classroom teachers, LSAs years 3–5; parents n=TBC | Semistructured interview | Postintervention | Thematic analysis |

Continued

**Table 1** Continued

| Domain | Research questions / process evaluation framework mapping | Informant/source of data | Methods | Timing of data collection | Analysis |
|---|---|---|---|---|---|
| CVI project intervention differentiation and usual practice Intervention schools n=4 Control schools n=3 | Was any contamination detected from intervention to control schools? | SENCO/head teacher n=7 (all schools) Head teacher/SENCO and classroom teachers, years 3–5 n=35 classes (control and intervention schools) | Semistructured interview Photographs of study designated viewpoints of each wall in years 3–5 classrooms | Follow-up data collection Prerandomisation and at follow-up data collection | Thematic analysis Validated University of Bristol 'clutter' measure Quantitative analysis |
| | Were there any changes in school SEND documentation over the course of the study that might indicate contamination? *Contamination* | SEND policy documents in all schools n=7 | Structured review of schools' SEND documentation | Pre-follow-up data collection | Content analysis |
| | What was different about the CVI project intervention when compared with usual practice within the school? *Differentiation, mediators to implementation of the intervention* | Head teacher/SENCO and classroom teachers, years 3–5 n=TBC | Semistructured Interview | Follow-up data collection | Thematic analysis |
| Acceptability of the intervention | Was the CVI project intervention acceptable to school staff and parents? *Acceptability: participants' responses and interactions with the intervention* | Intervention schools: SENCO/head teacher, classroom teacher/LSA years 3–5, parents n=TBC | Semistructured interview | Post-follow-up | Thematic analysis |
| | What were the experiences of staff and parents of taking part in the CVI project intervention? *Participant interactions with the CVI project intervention* | Intervention schools: SENCO/head teacher, classroom teachers, LSAs years 3–5; parents n=TBC | Semistructured interview | Post-follow-up | Thematic analysis |
| Sustainability of the intervention | Did schools/were schools intending to continue with intervention after the trial period? *Sustainability, participants' responses to the intervention, fidelity, quality of sustained use.* | SENCO/head teacher n=4 | Semistructured interview | Post-follow-up | Thematic analysis |
| | How was the intervention integrated into usual practice? | SENCO/head teachers, classroom teachers, years 3–5 n=TBC | Semistructured interview | Post-follow-up | Thematic analysis |
| | Was intervention integrated into SEND policies? *Mechanisms of sustainability of intervention* | SEND policy documents from intervention schools n=4 | Structured review of SEND policy documents | Prerandomisation Post-follow-up | Content analysis |
| | What factors made it more/less likely to be sustained/adopted into usual practice? *Barriers and facilitators/mediators to sustainability of the intervention* | SENCO/head teacher n=4 | Semistructured interview | Post-follow-up | Thematic analysis |

Continued

**Table 1** Continued

| Domain | Research questions *process evaluation framework mapping* | Informant/source of data | Methods | Timing of data collection | Analysis |
|---|---|---|---|---|---|
| Context Intervention schools n=4 | What were the experiences of paediatric ophthalmology clinics and orthoptists? *Participants responses to the intervention Barriers and facilitators to participation in the CVI project feasibility RCT* | Head orthoptist n=3 | Semistructured interview | Post-follow-up | Thematic analysis |
| | How did the referral process from schools to paediatric ophthalmology clinics work? *Fidelity, adaptations, reach, barriers and facilitators Contamination, unintended consequences* | Paediatric ophthalmology clinics n=3 SENCO/head teacher n=4 | Semistructured interview | Post-follow-up | Thematic analysis |
| | How many children were given study referral letters for their parents? Who were they referred to? | SENCO/head teacher n=4 | Records kept at each school Schools collate data | Throughout trial | Count and record |
| | What were the characteristics of the children who attended the paediatric ophthalmology clinic with a study referral letter? *Fidelity, dose, reach* | Paediatric ophthalmology clinics n=3 | CVI vision assessment Conducted by orthoptists | | Recorded on University REDCap database accessed via paediatric ophthalmology clinics |
| | How did contextual factors affect intervention implementation? (eg, funding pressures; reorganisation; academisation, Ofsted, SATS) *Implementation and mechanisms of impact* | SENCO/head teacher n=4 | Semistructured interview | Pre-follow-up data collection | Thematic analysis |
| Trial process Intervention and control schools n=7 | What did schools think about the recruitment process? *Trial methods, acceptability to participants. Participant responses to trial methods Barriers and facilitators* | Head teachers/SENCOs in control and intervention schools n=7 | Semistructured interview | Postbaseline data collection | Thematic analysis |
| | What was the initial recruitment rate? *Recruitment data* | Demographic data from publicly available government schools database | Extraction of demographic data | Postrecruitment | Descriptive data summaries |
| | What was the dropout rate of schools from the research? What reasons were given? *Participant responses and interactions with the trial processes* | Recorded as part of trial routine data collection | Quantitative data with some free text/coded responses for reasons | Ongoing throughout study period | Count and record |
| | How many parents withdrew their children from the study? What reasons were given? *Participant responses and interactions with the trial processes* | Recorded as part of trial routine data collection | Quantitative data with some free text/coded responses for reasons | Ongoing throughout study period | Count and record |
| | How did schools communicate with parents about trial? *Mechanisms of impact, fidelity, reach* | Examination of school documents/websites/newsletters | Structured review of documents | Ongoing throughout study period | Content analysis |
| | Acceptability of trial processes: how did schools respond to randomisation? Including the arm of the trial that they were allocated to? *Acceptability of trial processes, interactions of participants with trial processes* | SENCO/head teacher in intervention and control schools n=7 | Semistructured interview | Postrandomisation | Thematic analysis |

LSA, Learning Support Assistant; SENCO, special educational needs co-ordinator; SEND, special educational needs and disability; TBC, to be confirmed.

about knowledge of CVI (eg, 'I understand the impact of CVI in daily life') and respondents are asked to rate whether they feel each statement is true for them on a scale of 0–10.

### Qualitative interviews with school representative (all schools)

One key representative from each school (usually the SENCO or head teacher) will take part in a semistructured interview at baseline following randomisation (after the intervention has been delivered in intervention schools) and at follow-up. The initial interview will cover school ethos, experiences with data collection, trial processes (including randomisation) and any changes during the study period. Additional topics for intervention schools will include responses to the intervention pack, resources implemented, feedback from staff, monitoring and recording of changes. See online supplemental materials for topic guides.

### Qualitative interviews with school staff (intervention schools only)

School staff will be invited to take part in a semistructured interview postintervention implementation, to share their experiences of the CVI Project intervention. The interviews will explore reactions to the intervention materials, how they have been used, views and opinions on their use and usefulness, changes to provision, implementation of both universal and targeted strategies and any feedback on the data collection methods used in the trial. Interviews with staff will take place as the intervention is being used at the school.

### Qualitative interviews with parents (all schools)

Parents who returned 'consent to contact' forms will be contacted and invited to take part in a semistructured interview about their experiences with the study. Interviews will explore their experiences with the data collection, surveys and any thoughts they had about randomisation. Interviews will also cover any changes during the study period and what these can be attributed to and any referral processes and experiences with the paediatric ophthalmology clinics. Any positive, negative or unintended consequences will be discussed.

### Qualitative interviews with paediatric ophthalmology clinic staff

Paediatric ophthalmology clinic staff with involvement in the study will be invited to take part in a semistructured interview to explore their experiences with referrals from schools. The topics covered in these interviews will include the referral process, characteristics of children who are referred, communication between school, family and hospital unit.

### Classroom clutter (all schools)

We will ask schools to take photographs of each wall in the classrooms used by years 3–5 in each school, prerandomisation and again at follow-up. Photographs of the classrooms at the start and end of the study period will be compared and scored subjectively for the amount of clutter, by observers unaware of the school or time they were taken. We will also pilot using an adaptation of a visual clutter quantification system originally used in a study of how the visual characteristics of a natural scene affect bird and human search behaviour.[19] We anticipate that we will present the results of analyses using the photographs briefly in our results paper and separately in detail.

### School document analysis

We will investigate whether there any changes in school SEND documentation over the course of the study that might indicate contamination by conducting a structured review of all school's SEND policy documentation prerandomisation and at follow-up. They will be studied for any changes made during the study period. Changes found will be included as topics for discussion with SENCO and staff interviews.

### Analysis of qualitative data

With consent, the semistructured interviews will be audio-recorded and then transcribed verbatim, anonymised, checked for accuracy and imported into NVivo V.12.[20] Thematic analysis using a continuous comparative method will be used to identify recurrent ideas and patterns within the data.[21] Transcripts will be independently coded by two researchers to enhance rigour by cross-checking the coding strategy and early development of themes. During analysis, themes will be discussed among the team until a final consensus on the nature and phrasing of key themes and subthemes is reached. Anonymised quotes will be presented to illustrate the main themes.

### Analysis of quantitative data

Quantitative survey data will be presented as counts and percentages, as appropriate. String variables will be coded into categories. The CVI knowledge survey will be scored and comparisons made between baseline and follow-up. Survey data will be compared between intervention and control schools, but this will be exploratory and will be reported descriptively as the study is not powered to detect an effect size.[15]

Photographs of the classroom setting will be analysed using a clutter rating scale, developed at baseline with independent blind ratings by an academic not connected with the study, but with expertise in this area.

School policy documents will be subjected to content analysis by the study team, using a data extraction table to identify any differences between schools at the beginning and end of the study period.

### Ethics and HRA approval

Ethical approval for the study was obtained from the University of Bristol Faculty of Health Sciences Ethics Committee (FREC Ref: 89144). The study is registered on the International Standard Randomised Controlled Trial Number Registry (registration number: ISRCTN13762177). Approval from the Health Research Authority (HRA- Ref 19/HRA/6124) has been obtained.

Full details of consent procedures are described in our feasibility trial protocol paper (in preparation), but in brief: Head teachers agree to participate on behalf of the children and staff in his/her school, information sheets will be provided to school staff and parents, parents will be able to withdraw their child's data by completing a nonagreement form. Verbal agreement for child completed measures will also be sought and questionnaires will not be completed by any child who does not wish to do so. A link to the full privacy notice online[22] will be provided to all parents and school staff.

## Conclusions

We have presented a protocol for a detailed PE of a complex information sharing intervention to improve the mental well-being of primary school children with CVI. Development of an intervention to support children with CVI has been requested by parents,[23] and we have developed it with their input. This feasibility study will assess whether and how a future trial should establish effectiveness of the intervention with feedback from this PE. We have addressed key domains as per MRC guidance and have developed our PE research questions around these aims. Data gathered will help us to interpret the findings from the trial, but importantly, will help to refine the design, content and delivery of the intervention. We have also included data to improve trial processes and communication between schools and the research team in a future full-scale trial. Given that this is a PE for a feasibility trial, we will not be powered to interpret the primary outcome differences between intervention and control, and the data gathered as part of the PE will be limited in explaining any differences in the main feasibility study outcomes.

## TRIAL STATUS

Favourable opinion by the University of Bristol faculty ethics committee was obtained on 12 August 2019 (Ref: 89144). We completed recruitment and baseline data collection from participating schools in all areas. We have recruited seven schools to the feasibility trial and completed baseline trial data collection in February 2020. Four schools have been allocated to the intervention arm and three to control. Due to the COVID-19 global pandemic and school closures, we paused the study from March–September 2020. We have since obtained ethical approval for minor amendments relating to adjustments for a 'COVID-secure' restart to the study on 1 September 2020, with an extension to the study period until 30 April 2021. We will describe all COVID-19-related changes to the study in our final results paper, as these relate to the findings of our feasibility study.

**Acknowledgements** We are grateful to Mai Baquedano for her support with online consent and questionnaires, via the REDCap system. We are also grateful to our family and professionals advisory group members who provided valuable insight into recruitment and data collection strategies, and input into the intervention delivery.

**Contributors** CW led the development of the research including the design of the intervention and trial methods, with support from AP, TG and PS. RW provided data management and systems support. PS provided governance support. AP and TG led on the design of the process evaluation and associated research methodology. KB led the development of the health economics aspect of the process evaluation. All authors contributed to the final manuscript providing comments and text for the final submitted draft.

**Funding** University of Bristol provides Sponsorship for this study. The study is funded by the National Institute for Health Research (NIHR) Senior Research Fellowship ref: SRF-2015-08-005. The views expressed in this publication are those of the authors and not necessarily those of the NIHR, NHS or the UK Department of Health and Social Care.

**Competing interests** None declared.

**Patient consent for publication** Not required.

**Provenance and peer review** Not commissioned; externally peer reviewed.

**ORCID iDs**
Anna Pease http://orcid.org/0000-0002-3472-1047
Cathy Williams http://orcid.org/0000-0002-9133-2021

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
