## [Reviewer comments · BMJ Open]

ARTICLE DETAILS

TITLE (PROVISIONAL)	Improving outcomes for primary school children at risk of cerebral visual impairments (The CVI Project): study protocol for the process evaluation of a feasibility cluster-randomised controlled trial
AUTHORS	Pease, Anna; Goodenough, Trudy; Sinai, Parisa; Breheny, Katie; Watanabe, Rose; Williams, Cathy

VERSION 1 – REVIEW

REVIEWER	Nicola McDowell Institute of Education, Massey University, New Zealand
REVIEW RETURNED	20-Oct-2020

GENERAL COMMENTS	This is a very important and timely study. The focus on improving the well-being of children with CVIs is crucial to improve outcomes and services for children with this condition around the world. The interventions to be trialed are simple and easy to implement and could be easily replicated in classrooms around the world if proved effective. The authors have a deep understanding of the relevant research to build on the current knowledge around CVI. The research process is clearly outlined and explained. I am looking forward to seeing the results of this research.
---

REVIEWER	Lotfi Merabet Massachusetts Eye and Ear, Harvard Medical School
REVIEW RETURNED	27-Oct-2020

GENERAL COMMENTS	This protocol by Pease and coworkers outlines an investigation of how to improve outcomes for primary school children at risk for cerebral visual impairment (CVI), and centers on the effect of reducing environmental clutter. CVI is a condition that affects visual pathways and structures beyond the eye and thus many children with this condition have problems with higher order visual spatial processing. Current opinion suggests that simplifying (i.e. reducing clutter) the surrounding visual environment may help improve visual processing (e.g. longer attention and reduced visual fatigue) and thus translate to better school performance. However, robust and compelling evidence for this view is lacking. With CVI being the primary cause of pediatric visual impairment in developed countries, this study aims to address this gap in knowledge as well as a growing and important public health concern. In general, I found that the protocol well described and outlined. Certainly, the stratified and cluster-randomized approach brings a high level of strength to the study design. Outcomes are a mixture of quantitative and qualitative data. I also believe that grounding the approach in a logic model will also afford some structure in
--

	reasoning that will benefit the field overall. I have a few suggestions that may help strengthen an otherwise very important study. 1) I would suggest that more details be provided regarding the rating of environmental clutter. The authors cite the work of Xiao and Cuthill (2016) in terms of a clutter rating scale. However, as this was developed in a very different context (i.e. camouflage), I think it would be helpful if the authors provide more details of how they plan to adapt this scale to a school/classroom setting. 2) I am surprised that more medical information is not being collected on the CVI participants and also not using this information as part of the analysis. For example, what are the entering visual acuities and cause of CVI for the children? To me, these seem like very important variables (more specifically, effect modifiers) that would have an important impact on outcomes. There is a unique opportunity to gain insight on the effect of these variables with this study. 3) It may not be a journal requirement, but it would seem appropriate to develop a statistical analysis section. Specifically, what is the sample size justification? Was a power calculation carried out? Furthermore, as there appears to be a large amount of quantitative and qualitative data to be collected, what is the general statistical plan and corrections for multiple comparisons?
--	---

REVIEWER	Nofar Ben Itzhak Katholieke Universiteit Leuven, Belgium
REVIEW RETURNED	28-Oct-2020

GENERAL COMMENTS	This study has the potential for adding valuable information on CVI and improve awareness of CVI. Performing a feasibility study before a future full-scale trial is beneficial.
--

REVIEWER	Dr Swetha Sara Philip University of Queensland, Brisbane, Australia
REVIEW RETURNED	30-Oct-2020

GENERAL COMMENTS	Overall feedback: Thank you for the opportunity to review this paper. Cerebral Visual Impairment due to its varied clinical presentation, is often missed and therefore under reported. The findings in this paper, therefore, would be of interest to reader, however, I have comments to make. ABSTRACT: Introduction: First of all, looks like the authors are already conducting a study evaluating the feasibility of the interventions for CVI and this paper is a protocol evaluating how the intervention is implemented and provide interpretation. It is a bit confusing as one of the components 'feasibility of interventions' is to see how the interventions is implemented. Therefore to describe a protocol on implementation of interventions seems to a repetition. It would be nice for the authors to provide some more clarity on this. The dates of ongoing study needs to be mentioned. Methods and analysis: Line 27: what is SEND policy Strengths and limitations: Line 48: is Process evaluation or Process Evaluation (please check abstract) The page numbers have to be added in this manuscript, else giving
---

	a meaningful feedback becomes challenging. INTRODUCTION: Page 5, line 6: '.....collectively as....(CVI) or....(CVIs). It is better to avoid the term CVIs as it is already mentioned in the paper as brain related visual impairment, is a collective impairment, with some manifested and so given an alternate diagnosis. Page 5 line 44: We have...developed an intervention which.... It sounds incomplete, is it intervention of strategies or intervention tool.? Line 49: The development of the intervention..... As above the intervention needs to be specified...intervention of strategies or intervention tools. Line 53: logic model-see Fig 1 There is no logic model that I can view and if it is the image on Page 19 of this manuscript, I do not see any heading or Fig number. Pg 6 line 9: Reference 11 needs to be rechecked in context of what is mentioned in the text. Pg 6 beginning line 9 till line 13: The intervention also.....having CVI The same information is written in two different ways. Pg 6 line 42: how are the children getting the diagnosis of CVI? even before the study has started. If they have already been evaluated by the paediatric ophthalmologist and given the diagnosis of CVI, then why should they be again seen in the ophthalmology clinic. Please explain and provide better clarity. Pg 6 line 45: However, we also.....e.g. decluttering What does this sentence mean? Pg 6 line 48:the Primary outcome... Please correct it tothe primary outcome.... Pg 6 line 52: which are randomisation arms being referred to? Please give clarity to the randomisation process. Pg 7 line 3.....Feasibility.... Please change tofeasibility.... Line 40: Programme Differentiation and Usual Practice. Why should this contamination occur in control schools? Please explain. And if the contamination occurs would these schools be still considered as control schools Pg 10:Table 1: Fidelity, adaptation: Is there is a selection criteria for teachers and how many sessions of training do they have to undertake Table 1: Mechanisms of change of CVI project: 'What was the response of staff to the training?' This question is repeated at the end of the table section. Pg 14 line 27: when is the follow up? Time period? Pg 15 line 5: Clutter rating scale. Has this scale been described in the paper? Pg 16 DISCUSSION Subheading 'discussion' does not conform to the journal instruction.
--	--

VERSION 1 – AUTHOR RESPONSE

Comment	Response
Reviewers 1 and 3	We thank reviewers 1 and 3 for their supportive and kind comments

	regarding our manuscript.
Reviewer 2	
I would suggest that more details be provided regarding the rating of environmental clutter. The authors cite the work of Xiao and Cuthill (2016) in terms of a clutter rating scale. However, as this was developed in a very different context (i.e. camouflage), I think it would be helpful if the authors provide more details of how they plan to adapt this scale to a school/classroom setting.	Thank you, this was also raised by reviewer 4. We have amended this section and it now reads: “Photographs of the classrooms at the start and end of the study period will be compared and scored subjectively for the amount of clutter, by observers unaware of the time they were taken. We will also pilot using an adaptation of a visual clutter scoring system originally used in a study of how the visual characteristics of a natural scene affect bird and human search behaviour.”
I am surprised that more medical information is not being collected on the CVI participants and also not using this information as part of the analysis. For example, what are the entering visual acuities and cause of CVI for the children? To me, these seem like very important variables (more specifically, effect modifiers) that would have an important impact on outcomes. There is a unique opportunity to gain insight on the effect of these variables with this study.	The children in our study are unlikely to have been assessed for CVI, and in the real world it is highly unlikely that this would be available for every child. So whilst we agree with the reviewer that we would like this information, our aim is to investigate and understand an intervention that could be scaled up for widespread use if it is found to be helpful. However we may be able to access records relating to standardised vision tests carried out at school entry in future studies and we thank the reviewer for this suggestion
It may not be a journal requirement, but it would seem appropriate to develop a statistical analysis section. Specifically, what is the sample size justification? Was a power calculation carried out? Furthermore, as there appears to be a large amount of quantitative and qualitative data to be collected, what is the general statistical plan and corrections for multiple comparisons?	The plan for statistical analysis is described in detail in our main trial protocol, but we have addressed this point in our section: “Analysis of quantitative data” As we are not powered to detect meaningful differences between baseline and follow up (as is usual in feasibility studies), we will report the counts and percentage changes from the surveys. We will carry out exploratory comparisons of the data in the two arms to inform the power calculation for a future full trial. We discuss this in the accompanying trial methods protocol paper.
Reviewer 4	
First of all, looks like the authors are already conducting a study evaluating the feasibility of the interventions for CVI and this paper is a protocol evaluating how the intervention is implemented and provide interpretation. It is a bit confusing as one of the components ‘feasibility of interventions’ is to see how the interventions is implemented. Therefore to describe a protocol on implementation of interventions seems to a repetition. It would be nice for the authors to provide some more clarity on this.	Thank you and we think the word ‘feasibility’ is causing this issue. The process evaluation is nested within a feasibility trial to find out the best way to test the effectiveness of our intervention. The process evaluation itself will look at how the intervention was implemented (this includes how feasible and acceptable it was in practice) and the learning from this will be used to make any necessary modifications to the

	intervention to be tested in a full-scale trial.
The dates of ongoing study needs to be mentioned.	We have added in this detail to the second paragraph of the section titled "The CVI Project Feasibility Trial": Baseline and follow up data will be collected between September 2019 and February 2021
Methods and analysis: Line 27: what is SEND policy	We have clarified this acronym in the Abstract, where it first appears. It stands for Special Educational Needs and Disabilities.
Strengths and limitations: Line 48: is Process evaluation or Process Evaluation (please check abstract)	Thanks we feel that it does not require capitalisation or shortening and should be 'process evaluation', and have amended the abstract and text where appropriate.
The page numbers have to be added in this manuscript, else giving a meaningful feedback becomes challenging.	Journal guidelines do not require page numbers as page numbers are added by the journal to the pdf versions for reviewers.
Page 5, 1line 6: '.....collectively as....(CVI) or....(CVIs). It is better to avoid the term CVIs as it is already mentioned in the paper as brain related visual impairment, is a collective impairment, with some manifested and so given an alternate diagnosis.	We have updated the manuscript to 'CVI' as suggested.
Page 5 line 44: We have...developed an intervention which.... It sounds incomplete, is it intervention of strategies or intervention tool.?	We have changed it to: 'an intervention pack' as schools were given a pack of resources which aim to increase knowledge about CVI and provide strategies to help.
Line 49: The development of the intervention..... As above the intervention needs to be specified...intervention of strategies or intervention tools.	Added 'pack' here too, to make it clearer.
Line 53: logic model-see Fig 1 There is no logic model that I can view and if it is the image on Page 19 of this manuscript, I do not see any heading or Fig number.	The logic model is at the end of the pdf version. We will make sure it is uploaded as per journal guidance.
Pg 6 line 9: Reference 11 needs to be rechecked in context of what is mentioned in the text.	We have fixed this and updated reference 11 with the correct citation.
Pg 6 beginning line 9 till line 13: The intervention also.....having CVI The same information is written in two different ways.	We have deleted the repeated sentence.
Pg 6 line 42: how are the children getting the diagnosis of CVI? even before the study has started. If they have already been evaluated by the paediatric ophthalmologist and given the diagnosis of CVI, then why should they be again seen in the ophthalmology clinic. Please explain and provide	We have clarified this so it now reads: "In this trial, the primary outcome (child self-reported mental wellbeing) will be assessed in all children who are getting extra help, rather than only in children with confirmed CVI. We will not assess

better clarity.	every child in the study for CVI. This is because in real life this would not be possible and our aim is to evaluate an intervention that could be scaled up for widespread use, if found to be effective. Furthermore, if we examined all children at the study start to identify those who fitted a definition of CVI, or all those with CVI-related vision problems, it would then be unethical not to offer help to identified children and so we would not have a control group. In our earlier study(15), most children with CVI were found within the group getting extra help and these children are easily identifiable by schools (whereas children with confirmed CVI are not).”
Pg 6 line 45: However, we also.....e.g. decluttering What does this sentence mean?	We have clarified this so it now reads: “In addition, we also found some children with CVI-related vision problems who were not getting extra help and we hypothesize many children might benefit from some aspects of the intervention e.g. decluttering the classroom to make the visual environment less busy.”
Pg 6 line 48:the Primary outcome... Please correct it tothe primary outcome....	We have corrected this, thank you.
Pg 6 line 52: which are randomisation arms being referred to? Please give clarity to the randomisation process.	We have clarified this sentence to: “Schools will be told immediately whether they are allocated to the control group (and will be asked to carry on as normal for the school year) or the intervention group (where they will be invited to implement the CVI intervention pack, described above).”
Pg 7 line 3.....Feasibility.... Please change tofeasibility....	We have corrected this, thank you.
Line 40: Programme Differentiation and Usual Practice. Why should this contamination occur in control schools? Please explain. And if the contamination occurs would these schools be still considered as control schools	We have clarified this to: “Contamination from intervention to control schools is a potential risk, where schools in the control group make changes related to CVI after hearing about the intervention from a school in the intervention arm. We will seek to assess this with exploratory questions about how and why this occurred and the outcome for control schools.”
Pg 10:Table 1: Fidelity, adaptation: Is there is a selection criteria for teachers and how many sessions of training do they have to undertake	We have added in point 1. ‘Implementation of the intervention’: “We have made suggestions for how schools may wish to implement the intervention (including delivering the presentation at a staff meeting and sharing the resources with class teachers), however we have not been

	prescriptive about this. Early discussions with teachers in our advisory groups suggested that individual schools would prefer to find their own ways to implement the intervention pack, and that autonomy over this process would be appreciated.”
Table 1: Mechanisms of change of CVI project: 'What was the response of staff to the training?' This question is repeated at the end of the table section.	We have deleted the duplicate row, thank you.
Pg 14 line 27: when is the follow up? Time period?	We have included the time period, as above.
Pg 15 line 5: Clutter rating scale. Has this scale been described in the paper?	Thank you, reviewer 2 also raised this and we have amended this section and it now reads: “Photographs of the classrooms at the start and end of the study period will be compared and scored subjectively for the amount of clutter, by observers unaware of the time they were taken. We will also pilot using an adaptation of a visual clutter scoring system originally used in a study of how the visual characteristics of a natural scene affect bird and human search behaviour.”
Pg 16 DISCUSSION Subheading 'discussion' does not conform to the journal instruction.	We have changed this to 'Conclusions' in line with author guidelines, thank you.
Patient and Public Involvement: - We have implemented an additional requirement to all articles to include 'Patient and Public Involvement' statement within the main text of your main document. Please refer below for more information regarding this new instruction: Authors must include a statement in the methods section of the manuscript under the sub-heading 'Patient and Public Involvement'. This should provide a brief response to the following questions: How was the development of the research question and outcome measures informed by patients' priorities, experience, and preferences? How did you involve patients in the design of this study? Were patients involved in the recruitment to and conduct of the study? How will the results be disseminated to study participants? For randomised controlled trials, was the burden of the intervention assessed by patients themselves? Patient advisers should also be thanked in	We have added: “Patient and Public Involvement We have spoken with two groups of parents whose children have special educational needs and/or CVI. They have been consulted at every stage of the research so far and provided guidance on which measures to use with parents, and which topics to include in qualitative interviews. They advised us on the need to raise awareness in schools about CVI, and we changed our intervention pack to include specific resources to address this. They were clear that teachers should be reimbursed for their time to take part in the study and that we should outline the potential benefits of identifying CVI whereby the right strategies can be put in place, freeing up teacher time not spent trying ineffective interventions. They also advised us that schools might appreciate the benefit of being able to

the contributorship statement/acknowledgements. If patients and or public were not involved please state this.	demonstrate to OFSTED (Office for Standards in Education) inspectors that they were implementing strategies to support students who may be struggling with their learning.” We have already thanked our parent advisors in the acknowledgements
---	---

VERSION 2 – REVIEW

REVIEWER	Swetha Philip The University of Queensland
REVIEW RETURNED	23-Jan-2021

GENERAL COMMENTS	Introduction: Thanks for clarifying regarding the feasibility and process evaluation. 1. My concern is writing this protocol paper on the feasibility and acceptance of intervention does not quite fit the order when the study regarding the feasibility of the intervention is already being conducted/ underway as mentioned in the manuscript. The protocol paper should have been published first before the study is underway. 2. 'This paper describes the protocol for an trial outcomes' Is the interpretation of the feasibility trial outcome for this study or for the study trials in the future? MAIN TEXT: 3. Page 4, 1line 6: '.....collectively as....(CVI) or....(CVIs). As suggested in the previous review to avoid the term 'CVIs'. The correction has not been made in the revised manuscript. 4. Page 4 line 11-12:maybe misinterpreted by teachers....difficulties. May be worth mentioning the 'other difficulties' the child is labelled with so that the readers are made aware that behavioural/learning/intellectual difficulties may all be different manifestation of CVI. 5. Page 4 line 48:and we changed our intervention pack to include.... This is a protocol paper. The protocol has been developed including intervention pack with the input of different advisory groups and experience of the investigators. All the changes/ modifications are part of the original protocol. When this statement is made ' we changed our intervention pack' it seems to the readers that the intervention pack was changed midway when the protocol was being executed for feasibility and acceptability study. If that is so, then it cannot be part of the protocol paper but part of another paper analysing the results of the protocol. 6. Page 5 Line 14: logic model-see Fig 1 I am still unable to view the logic model figure in the manuscript. If the figure on Page 19 of the PDF format is the figure of logical model, there is no heading to it. 7. Page 5 line 28: Reference 12: As suggested in the previous review, the reference Bartimeus is not listed in the reference list. This has not been corrected in this revised manuscript.
--

	8. Page 5 line 56:data will be collected between September 2019 and February 2021 on.... Is this February 2021 or February 2020? Please clarify as this date is different from what is mentioned under the heading 'Trial Status'. 9. Page 6 first paragraph: The process of randomisation and rationale for it is not clear unless one reads the next two paragraphs. This paragraph is needs to be modified such that the study is easily understood by the reader. 10. Page 6 line 14:..... the primary outcome will be.....in a secondary outcome analysis. Please clarify this statement. Please clarify what is the secondary outcome that would be analysed. 11. Page 6 line 39: ...conceptual framework for the PE. This abbreviation has not been explained in the manuscript. 12. Page 14 line 7-8:The survey includes 6 questions about.... Are these survey questions baseline questions? 13. Page 15 line 6:..... and scored subjectively for the.... How will this score be standardised across the study cohort? 14. Page 15 line 7:.....will also pilot using an adaptation..... Is this another clutter scoring system that is being used in this study? The clutter rating system used by Xiao et al 2016, incorporates colour and complexity of the scene. Are the authors checking for colour blindness in study participants? There is no scoring system provided by Xiao et al in their article. How do the authors of this study plan to adapt the scoring (grading)? 15. Page 15 line 40:using a clutter rating scale, developed at.... Has this been developed for the protocol or yet to be developed? There is no clutter rating scale that can be viewed along with this manuscript if it has been developed.
--	--

VERSION 2 – AUTHOR RESPONSE

Reviewer: 4

Very many thanks for the opportunity to make all the requested corrections and improve our manuscript.

A point-by-point response is below:

ABSTRACT:

Introduction:

Thanks for clarifying regarding the feasibility and process evaluation.

1. My concern is writing this protocol paper on the feasibility and acceptance of intervention does not quite fit the order when the study regarding the feasibility of the intervention is already being conducted/ underway as mentioned in the manuscript. The protocol paper should have been published first before the study is underway.

We submitted this manuscript to BMJ as it says in the Instructions to Authors that ongoing, but not completed studies are eligible for Protocol papers:

“Protocol manuscripts should report planned or ongoing research studies. If data collection is complete, we will not consider the manuscript. We encourage the submission of protocol manuscripts at an early stage of the study. Protocols nearing completion of data collection will

be treated on a case by case basis and the final decision on whether to consider a protocol for publication will rest with the Editor.”

At the time of submission (Sept 2021), we had just restarted after a pause caused by the pandemic and were at an early stage in the intended process, having just completed baseline assessments immediately prior to lockdown.

We agree that Protocols should be published as early as possible and have been hoping to publish this as soon as we could, once we knew the study was able to proceed. We are happy to be guided by the Editors as to whether our paper is suitable for inclusion in BMJ Open.

2. *‘This paper describes the protocol for an trial outcomes’
Is the interpretation of the feasibility trial outcome for this study or for the study trials in the future?*

The aim of this Process Evaluation is to provide context for the results of this ongoing feasibility study, not a future one.

MAIN TEXT:

3. *Page 4, line 6: ‘.....collectively as....(CVI) or....(CVIs).
As suggested in the previous review to avoid the term ‘CVIs’. The correction has not been made in the revised manuscript.*

Thank you for pointing this out and we have now removed the term CVIs .

4. *Page 4 line 11-12:maybe misinterpreted by teachers....difficulties.
May be worth mentioning the ‘other difficulties’ the child is labelled with so that the readers are made aware that behavioural/learning/intellectual difficulties may all be different manifestation of CVI.*

We have amended this line to

“.....behavioural, learning or intellectual difficulties....”

5. *Page 4 line 48:and we changed our intervention pack to include....
This is a protocol paper. The protocol has been developed including intervention pack with the input of different advisory groups and experience of the investigators. All the changes/ modifications are part of the original protocol.
When this statement is made ‘ we changed our intervention pack’ it seems to the readers that the intervention pack was changed midway when the protocol was being executed for feasibility and acceptability study. If that is so, then it cannot be part of the protocol paper but part of another paper analysing the results of the protocol.*

We were describing the development process for the intervention, prior to the study commencing. To make this more clear we have amended the line to:

“During the study design phase, the parents advised us on the need to raise awareness in schools about CVI, and we changed our draft intervention pack to include specific resources to address this.”

6. *Page 5 Line 14: logic model-see Fig 1
I am still unable to view the logic model figure in the manuscript.
If the figure on Page 19 of the PDF format is the figure of logical model, there is no heading to it.*

The Logic Model is the figure on p20 of the pdf as we have downloaded it. The words Logic Model are in the filename and the title is given in the main document at P6 line 42

7. *Page 5 line 28: Reference 12: As suggested in the previous review, the reference Bartimeus is not listed in the reference list.
This has not been corrected in this revised manuscript.*

Thank you have now corrected this reference

8. *Page 5 line 56:data will be collected between September 2019 and February 2021 on.... Is this February 2021 or February 2020? Please clarify as this date is different from what is mentioned under the heading 'Trial Status'.*

Thank you we agree this was confusing. We have removed the dates from the text and left all the description of the timing for the Trial Status section

9. *Page 6 first paragraph: The process of randomisation and rationale for it is not clear unless one reads the next two paragraphs. This paragraph is needs to be modified such that the study is easily understood by the reader.*

We have reordered the paragraph to make our reasoning more clear

10. *Page 6 line 14:..... the primary outcome will be.....in a secondary outcome analysis. Please clarify this statement. Please clarify what is the secondary outcome that would be analysed.*

We have rephrased as

“the primary outcome of child-reported quality of life will be assessed in the children getting extra help, but we will include all children in an additional analysis using the same outcome.”

11. *Page 6 line 39: ...conceptual framework for the PE. This abbreviation has not been explained in the manuscript.*

Thank you - we have corrected this and inserted the words Process Evaluation before the initials PE

12. *Page 14 line 7-8:The survey includes 6 questions about.... Are these survey questions baseline questions?*

These survey questions refer to the implementation of the intervention and will be asked during the intervention period. We have added “during the intervention period” to the preceding sentence for clarity.

13. *Page 15 line 6:..... and scored subjectively for the.... How will this score be standardised across the study cohort?*

We will ask one or more observers to score all pictures, without knowledge of which school they were taken in.

14. *Page 15 line 7:.....will also pilot using an adaptation..... Is this another clutter scoring system that is being used in this study? The clutter rating system used by Xiao et al 2016, incorporates colour and complexity of the scene. Are the authors checking for colour blindness in study participants? There is no scoring system provided by Xiao et al in their article. How do the authors of this study plan to adapt the scoring (grading)?*

Thank you- we have now clarified the text to explain our plans better. We have not collected any data on trial participants colour vision and will include this when we write up the results of the pilot adaption of the Xiao methods.

“We will also pilot using an adaptation of a visual clutter quantification system originally used in a study of how the visual characteristics of a natural scene affect bird and human search

behaviour{Xiao, 2016 #3139}. We anticipate that we will present the results of analyses using the photographs briefly in our results paper and separately in full.”

15. Page 15 line 40:using a clutter rating scale, developed at....
 Has this been developed for the protocol or yet to be developed? There is no clutter rating scale that can be viewed along with this manuscript if it has been developed.

The clutter rating scale is yet to be developed therefore is not presented

Reviewer: 4

Competing interests of Reviewer: None declared

VERSION 3 – REVIEW

REVIEWER	Swetha Philip The University of Queensland
REVIEW RETURNED	07-Mar-2021

GENERAL COMMENTS	Overall feedback: Thank you for the revising this paper and addressing the points raised in the previous reviews. The manuscript is much more easier to understand now. There are some punction/reference errors which to be corrected and some sentences need some more clarity as listed below: Pg 6, line7. However, in our...1(10), we.... Please correct that. Pg 6, line 7. I (10), we found that most children with CVI.....are easily identifiable by schools (whereas children with confirmed CVI are not) The information in that sentence seems contraindicating....please clarify/give clarity to what this sentence is meant to convey. Pg 6, line 12. We will not assess.... (10) In addition.... Please correct that. Pg 6,line 16.We will therefore collect data for all children.....but we will include all children in an additional analysis using the same outcome. This statement is still not clear in what it intends to convey. Please modify to provide more clarity.
---

VERSION 3 – AUTHOR RESPONSE

Pg 6, line7. However, in our...1(10), we....
Please correct that.
Done

Pg 6, line 7. I (10), we found that most children with CVI.....are easily identifiable by schools (whereas children with confirmed CVI are not)
The information in that sentence seems contraindicating....please clarify/give clarity to what this sentence is meant to convey.
This is discussed further down that paragraph and so we have removed the text in brackets.

Pg 6, line 12. We will not assess.... (10) In addition....
Please correct that.
Done.

Pg 6,line 16.We will therefore collect data for all children.....but we will include all children in an additional analysis using the same outcome.

This statement is still not clear in what it intends to convey. Please modify to provide more clarity.

Thanks, we have updated the text to "We will therefore collect data for *all* children in years 3-5. The primary outcome of child-reported quality of life will be assessed in the children getting extra help, and we will include all children in an additional (secondary) analysis using the same outcome."